# Prospective Study of Attachment as a Predictor of Binge Eating, Emotional Eating and Weight Loss Two Years after Bariatric Surgery

**DOI:** 10.3390/nu11071625

**Published:** 2019-07-17

**Authors:** Samantha E. Leung, Susan Wnuk, Timothy Jackson, Stephanie E. Cassin, Raed Hawa, Sanjeev Sockalingam

**Affiliations:** 1Centre for Mental Health, University Health Network, Toronto, ON M5G 2C4, Canada; 2Bariatric Surgery Program, Toronto Western Hospital, Toronto, ON M5T 2S8, Canada; 3Department of Psychiatry, University of Toronto, Toronto, ON M5T 1R8, Canada; 4Division of General Surgery, University Health Network, University of Toronto, Toronto, ON M5T 2S8, Canada; 5Department of Psychology, Ryerson University, Toronto, ON M5B 2K3, Canada; 6Centre for Addiction and Mental Health, Toronto, ON M5S 2S1, Canada

**Keywords:** bariatric surgery, attachment theory, disordered eating, binge eating, predictors

## Abstract

Bariatric surgery remains the most effective treatment for severe obesity, though post-surgical outcomes are variable with respect to long-term weight loss and eating-related psychopathology. Attachment style is an important variable affecting eating psychopathology among individuals with obesity. To date, studies examining eating psychopathology and attachment style in bariatric surgery populations have been limited to pre-surgery samples and cross-sectional study design. The current prospective study sought to determine whether attachment insecurity is associated with binge eating, emotional eating, and weight loss outcomes at 2-years post-surgery. Patients (*n* = 108) completed questionnaires on attachment style (ECR-16), binge eating (BES), emotional eating (EES), depression (PHQ-9), and anxiety (GAD-7). Multivariate linear regression analyses were conducted to examine the association between attachment insecurity and 2-years post-surgery disordered eating and percent total weight loss. Female gender was found to be a significant predictor of binge eating (*p* = 0.007) and emotional eating (*p* = 0.023) at 2-years post-surgery. Avoidant attachment (*p* = 0.009) was also found to be a significant predictor of binge eating at 2-years post-surgery. To our knowledge, this study is the first to explore attachment style as a predictor of long-term post-operative eating pathology and weight outcomes in bariatric surgery patients.

## 1. Introduction

Bariatric surgery remains the most effective treatment for severe obesity with respect to weight loss and resolution of obesity-related comorbidities, including diabetes [1,2]. However, post-surgical outcomes are variable with respect to long-term weight loss and eating-related psychopathology [3], and systematic reviews have identified a multitude of factors related to weight regain following bariatric surgery including psychiatric illnesses [4]. Post-operative recurrence of maladaptive eating behaviors, such as grazing or binge eating, have been associated with increased weight regain after bariatric surgery [5]. Data from the Longitudinal Assessment of Bariatric Surgery (LABS)-3 Psychosocial Study showed that increased hunger and greater eating psychopathology post-surgery were associated with less weight loss 3-years post-surgery [6]. Moreover, a recent Canadian study showed that higher scores on the binge eating scale (BES) at one-year post-surgery were associated with a lower percentage of total weight loss (%TWL) at two-years post-surgery [7].

An important variable affecting eating psychopathology among individuals with obesity is attachment style, which has also been used as a more specific marker for social support. Attachment theory posits that one’s attachment (relationship) style is formed in early childhood based on a child’s bond with his or her parent, thus forming an internalized working model of the self and others. This working model is applied to future relationships including healthcare relationships and can predispose an individual to a predictive response in these relationships and during illness events [8]. Moreover, attachment styles are durable over time with research showing a 72% stability rate from infancy into adulthood [9].

Attachment theory has been used to help explain individual differences in emotion regulation in a range of medical conditions such as diabetes [10], hepatitis C [11], chronic pain [12], and obesity [13]. An avoidant attachment style consists of over-regulation of emotions and avoiding perceptions of any negative self-views or personal weaknesses through self-reliance and independence, whereas an anxious attachment style consists of difficulties with emotional regulation in response to perceived interpersonal abandonment or rejection. Both of these insecure attachment styles have been associated with poor vitamin adherence and appointment attendance following bariatric surgery [14,15]. Studies examining the association between insecure attachment style and weight loss in bariatric surgery patients have yielded inconclusive results [16,17]; however, one study found that poor dietary adherence mediated the relationship between anxious attachment style and weight loss [17]. Data from non-bariatric patient samples have shown greater eating psychopathology among patients with insecure attachment style, which could also be relevant to eating psychopathology outcomes in bariatric surgery samples [18,19,20].

To date, studies examining attachment style and eating psychopathology in bariatric surgery populations have been limited to pre-surgery samples and cross-sectional study designs. For example, higher attachment anxiety has been associated with greater emotional eating [21] and binge eating symptoms [22] among bariatric surgery candidates. In both studies, emotion dysregulation was a significant mediator of the relationship between attachment insecurity and eating psychopathology, specifically emotional eating and binge eating symptoms, and this finding has been replicated in other non-clinical samples [23]. Previous studies have suggested a relationship between attachment insecurity and higher body mass index; however, this relationship is thought to be mediated by disinhibited eating in bariatric surgery candidates and recipients [24].

The primary objective of the current prospective study was to examine whether attachment insecurity is associated with eating psychopathology, specifically binge eating and emotional eating symptoms, at 2-years post-surgery. The secondary objective was to examine the association between attachment insecurity and %TWL 2-years post-surgery. Insecure attachment style was hypothesized to predict disordered eating symptoms and poorer weight loss outcomes 2 years post-surgery.

## 2. Materials and Methods

### 2.1. Study Setting

Patients were recruited between 2011 and 2014 as part of the Toronto Bariatric Surgery Psychosocial (Bari-PSYCH) study, which was conducted at the Toronto Bariatric Surgery Centre of Excellence (TBSCE) [25] and approved by the University Health Network Research Ethics Board. All patients gave their informed consent for inclusion before they participated in the study. The pre-surgical assessment and the post-surgical follow-up process in the TBSCE has previously been described in the literature and consists of an interprofessional and integrated approach to care [26,27,28]. Participants were between the ages of 18 and 65 years and had a pre-operative body mass index (BMI) >40 kg/m^2^ or BMI ≥35 kg/m^2^ with at least one obesity-related comorbidity. Patients were referred provincially for a Roux-en-Y Gastric Bypass through a centralized referral system, and some patients received a sleeve gastrectomy if surgically indicated (e.g., a history of previous abdominal surgeries resulting in extensive adhesions and/or distorted anatomy).

### 2.2. Study Procedures and Measures

The study received research ethics approval from the University Health Network Research Ethics Board. A total of 108 patients completed the specific study questionnaires and were included in the analyses. Pre-surgery demographic data were collected during clinical assessment and included sex, age, ethnicity, and marital status. Height and weight were measured by a clinician pre-surgery, and weight was subsequently measured at routine annual post-surgery follow-up appointments. Percent total weight loss (%TWL) was calculated by dividing the difference in weight at the specified time point by the original weight and then multiplying by 100.

Participants completed questionnaires at pre-surgery and then annually post-surgery as part of the Toronto Bari-PSYCH study protocol. The following questionnaires were included in the analysis of attachment style and to adjust for related psychosocial variables.

Attachment (relationship) style was assessed using the Experiences in Close Relationships scale (ECR-16), a 16-item scale that has been validated against the longer ECR-32 scale [29]. This measure yields anxious (ECR-16_anx_) and avoidant (ECR-16_avoid_) relationship style scores based upon the scoring of eight items related to each style, ranging from 8 to 56 for each domain. Higher scores on ECR-16_anx_ and ECR-16_avoid_ indicate higher attachment insecurity in these respective areas. The ECR-16 has been used to examine attachment style as a predictor of bariatric surgery follow-up appointment attendance, and also to examine the association between attachment style and health-related quality of life [14,30].

Binge eating symptoms were assessed using the binge eating scale (BES), a 16-item self-report questionnaire [31] that generates a total score ranging from 0 to 46. Binge eating symptoms are considered to be severe with scores at or above 27, moderate with scores between 18 and 26, and minimal or absent symptoms with score below 17. The BES has been validated in bariatric surgery populations [32].

Emotional eating symptoms were assessed using the emotional eating scale (EES), which is a 25-item self-report questionnaire [33]. The EES assesses patients’ propensity to eat in response to negative emotions such as anger (EES-anger), anxiety (EES-anxiety), or low mood states (EES-depression). Each item is rated on a 5-point Likert scale and a total EES score is calculated by summing the sub-scale scores. Total scores on the EES range from 0 to 100, with higher scores indicating greater emotional eating. The EES has been shown to have good construct validity and internal consistency and has been used in the bariatric population [33,34].

Depressive symptoms were assessed using the Patient Health Questionnaire-9 item (PHQ-9) [35]. Each item is scored from 0 (“not at all”) to 3 (“nearly every day”). PHQ-9 total scores range from 0 to 27, with higher scores indicating greater depressive symptoms. The PHQ-9 has been used in several studies assessing depressive symptoms in pre- and post-surgical bariatric patients [25,36,37].

Anxiety symptoms were assessed using the Generalized Anxiety Disorder-7 item (GAD-7). Each item is scored from 0 (“not at all”) to 3 (“nearly every day”). GAD-7 total scores range from 0 to 21. A score of 10 or greater represents an acceptable cut-off point for identifying generalized anxiety disorder, and also represents a sensitivity and specificity of 0.89 and 0.82 respectively [38,39]. The GAD-7 is validated to assess anxiety symptoms in primary care patients and specialized patient populations, including bariatric surgery patients [25,36,38].

### 2.3. Data Analysis

The data were analyzed using the Statistical Package for Social Sciences Statistics version 23.0 (SPSS 23.0, Armonk, NY, USA). Three multivariate linear regression models were used to identify whether attachment insecurity was a predictor of binge eating, emotional eating, and %TWL at 2-years post-surgery. Covariates for the linear regression models included age, gender, ethnicity, and pre-surgery relationship status, GAD-7, PHQ-9, binge eating, emotional eating, and weight. Statistical significance was determined with *p* < 0.05.

## 3. Results

Complete data were available for 108 patients. Patients had a mean age of 46.21 years (SD = 9.73), and the majority were female (80.6%), Caucasian (87.0%), and in a relationship (62.1%) (Table 1). Of the total sample, 99 patients received Roux-en-Y gastric bypass surgery and nine patients received sleeve gastrectomy. Mean pre-surgery BMI was 48.3 kg/m^2^ (SD = 8.72). These data are comparable to the larger Toronto Bari-PSYCH population, as those with incomplete data had a mean pre-surgery BMI of 48.5 kg/m^2^ (SD = 8.83), mean age of 44.98 years (SD = 11.07), and the majority were female (78.2%), Caucasian (82.9%), and in a relationship (61.2%).

Mean scores for BES, EES, PHQ-9, and GAD-7 were calculated at baseline and 2-years post-surgery. Over this time period, BES scores changed from 15.86 (SD = 7.27) to 7.21 (SD = 6.78), EES scores from 63.7 (SD = 22.06) to 40.2 (SD = 16.01), PHQ-9 scores from 9.96 (SD = 6.19) to 4.51 (SD = 4.620), and GAD-7 scores from 5.68 (SD = 5.42) to 3.10 (SD = 3.79).

The first multivariate linear regression model showed that gender (i.e., being female) and avoidant attachment style were significant predictors of BES scores at 2-years post-surgery (*r* = −2.670; *p* = 0.008 and *r* = 2.582; *p =* 0.010, respectively) (Table 2).

A secondary multivariate linear regression model showed that gender (i.e., being female) was a significant predictor of EES scores at 2-years post-surgery (*r* = −2.329; *p* = 0.021) (Table 3).

Neither anxious nor avoidant attachment styles were significant predictors of EES at 2-years post-surgery. No variables in the third multivariate linear regression model were significant predictors of %TWL at 2-years post-surgery (Table 4). Tests for multi-collinearity indicated a very low level of multi-collinearity among the regression models, with variance inflation factors less than four and tolerance values approaching one.

The relationship between attachment style and BES scores at 2-years post-surgery, EES scores at 2-years post-surgery, and %TWL are depicted in Figure 1a,b, Figure 2a,b and Figure 3a,b, respectively.

## 4. Discussion

To our knowledge, this study is the first to explore attachment style as a predictor of long-term post-operative eating pathology and weight outcomes in bariatric surgery patients. An avoidant attachment style was identified in this study as a significant predictor of binge eating at 2-years post-surgery. This finding is in contrast to a previous study, which found that avoidant attachment had a non-significant association with binge eating [20]. However, this previous study was a cross-sectional analysis examining pre-operative patients, and it is therefore possible that binge eating symptoms in those with an avoidant attachment style do not arise until after surgery. Although attachment avoidance is associated with a tendency towards over-control of emotion, previous research has shown that binge eating and attachment avoidance are both symptoms and manifestations which are characteristic of emotion dysregulation [22]. It is possible that increased emotion dysregulation after bariatric surgery may partially account for the significant association in our sample between attachment avoidance and binge eating symptoms.

Another potential explanation for the relationship between attachment avoidance and binge eating symptoms post-bariatric surgery is the role of non-adherence. An avoidant attachment style has been found to predict non-attendance at post-bariatric surgery follow-up appointments [22] and has also been associated with higher rates of patient drop-out from group cognitive behavioral therapy (CBT) for binge eating disorder [40]. It is possible that individuals with avoidant attachment may be non-adherent to their nutrition regimen following surgery and may not seek help, which could increase the risk of disordered eating and loss of control over eating, resulting in binge eating symptoms. However, further research on the role of emotion dysregulation and post-surgery nutrition adherence is needed to further understand these study findings.

Results from this study did not find an insecure attachment style to be a predictor of emotional eating or %TWL at 2-years post-surgery. Our findings are in line with the limited published research on attachment style and weight loss after bariatric surgery [16]. In general, prospective studies have not been able to identify pre-operative predictors of post-operative weight outcomes. Given that this study showed that pre-operative avoidant attachment predicted 2-year BES scores and there is empirical evidence from other studies to suggest that post-operative binge eating predicts poorer weight outcomes [7], it could be speculated that avoidant attachment may also predict longer term weight outcomes (i.e., at 3-years post-surgery). A recently published study looking at attachment orientations, eating behaviors, and obesity in the general population found that there was a significant, indirect effect between disorganized attachments and BMI via uncontrolled eating [41]. This mediation effect supports our hypothesis that attachment styles may indeed predict weight outcomes, albeit later on in the post-operative phase, though further research to delineate this association is warranted.

Our findings on emotional eating differ from previous studies that have shown significant relationships between attachment anxiety and emotional eating [21,23]. Our study may not have had a large enough sample size to identify a significant association between emotional eating and insecure attachment style. Further, emotional eating symptoms have been associated with fewer symptoms of psychopathology and loss of control over eating than binge eating symptoms in non-bariatric surgery patient samples [42]. Therefore, it is possible that binge eating is more likely than emotional eating to be associated with insecure attachment styles, which have been associated with greater distress and psychopathology.

The relationship between attachment avoidance and increased binge eating symptoms after bariatric surgery provides a rationale for psychosocial interventions to better support patients with an avoidant attachment style. Previous literature suggests that individuals with an avoidant attachment style may be more likely to engage in follow-up care and psychosocial interventions if they have more autonomy and their care is limited to a few key team members with whom they have formed a therapeutic alliance [43]. These team members may be perceived as trustworthy and increase patient receptiveness to engage in care and additional support over time. In addition, use of personalized and accessible psychosocial interventions to support post-operative coping and care, such as online interventions and tele-videoconferencing support, may improve the autonomy for patients with high attachment avoidance [6,43]. These specific approaches to psychosocial care and support could help increase patient engagement in after-care following bariatric surgery and potentially mitigate against the emergence of binge eating symptoms beyond the first year after surgery.

The study findings should be considered within the following limitations. First, the study sample size was limited to a subset of the Toronto Bari-PSYCH cohort that completed all necessary questionnaires. Additionally, the sample was predominantly female and Caucasian, though it is worth noting that these demographics are representative of the patient population within most bariatric programs and the 108 patients used in this study had baseline demographic data that were comparable to the overall Toronto Bari-PSYCH cohort. Moreover, the data were collected from patients in a single bariatric surgery center in Canada and thus require replication in other settings. Lastly, patients’ medication use was not looked at in this study, although antidepressants have been shown to be the most common medication used by bariatric surgery patients [44,45]. As data on the effects of antidepressants on attachment style remain limited in this population, further research is warranted in this area.

## 5. Conclusions

In summary, this study adds to the limited but growing literature on the influence of attachment style on binge eating symptoms. Attachment avoidance was found to be a significant predictor of binge eating at 2-years post-surgery, highlighting the need for recognition of patients’ attachment style and providing preventative treatment earlier on in the post-operative phase. Future research using a mediation model is warranted to delineate whether insecure attachment is a mediator of long-term emotional eating and weight outcomes.

## Figures and Tables

**Figure 1 nutrients-11-01625-f001:**
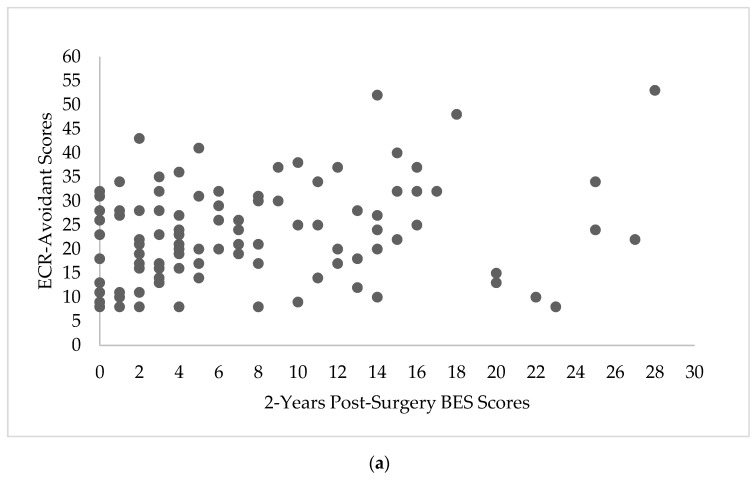
Scatterplots depicting the relationship between 2-year Binge Eating Scale (BES) scores and Experiences in Close Relationship Scale (ECR) scores. (**a**) Scatterplot depicting the relationship between 2-years post-surgery BES scores and ECR-Avoidant scores (*n* = 108). (**b**) Scatterplot depicting the relationship between 2-years post-surgery BES scores and ECR-Anxious scores (*n* = 108).

**Figure 2 nutrients-11-01625-f002:**
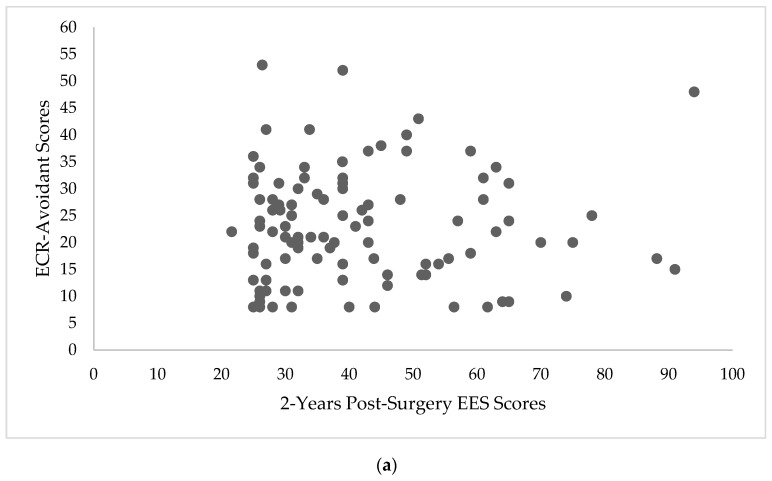
Scatterplots depicting the relationship between 2-year Emotional Eating Scale (EES) scores and Experiences in Close Relationship Scale (ECR) scores. (**a**) Scatterplot depicting the relationship between 2-years post-surgery EES scores and ECR-Avoidant scores (*n* = 108). (**b**) Scatterplot depicting the relationship between 2-years post-surgery EES scores and ECR-Anxious scores (*n* = 108).

**Figure 3 nutrients-11-01625-f003:**
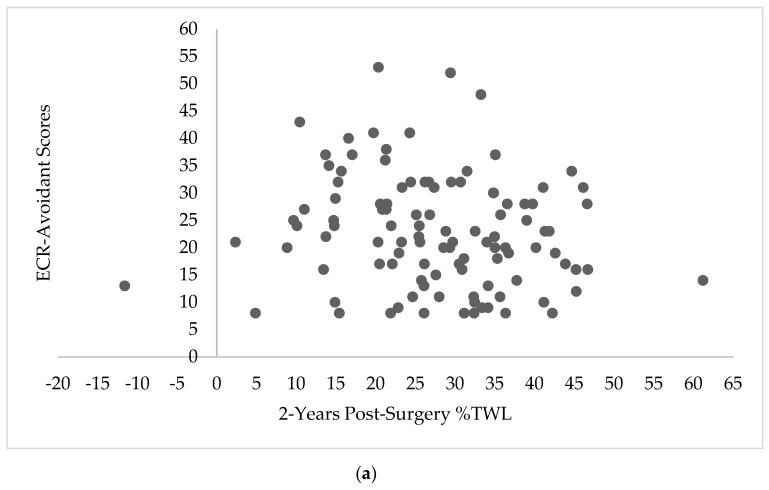
Scatterplots depicting the relationship between 2-year percent total weight loss (%TWL) and Experiences in Close Relationship Scale (ECR) scores. (**a**) Scatterplot depicting the relationship between 2-years post-surgery %TWL and ECR-avoidant scores (*n* = 108). (**b**) Scatterplot depicting the relationship between 2-years post-surgery %TWL and ECR-anxious scores (*n* = 108).

**Table 1 nutrients-11-01625-t001:** Participant demographics (*n* = 108).

Variable	M (SD) or n (%)
Age (years)	46.21 (9.728)
Gender (female)	87 (80.6%)
Race/Ethnicity	
Black	5 (4.6%)
East Asian	2 (1.9%)
Latin American	3 (2.8%)
South East Asian	2 (1.9%)
White (Caucasian)	94 (87.0%)
Relationship Status	
Married	56 (51.9%)
Common-Law	11 (10.2%)
Divorced/Separated	12 (11.1%)
Single	29 (26.9%)

**Table 2 nutrients-11-01625-t002:** Multivariate linear regression analysis with outcome variable 2-years post-surgery Binge Eating Scale (BES) scores (*n* = 108).

Variable	*r*	*p*
Age	−0.847	0.397
Gender (Female)	−2.670	0.008 *
Race/Ethnicity (Caucasian)	0.445	0.656
Relationship Status (In a Relationship)	−0.252	0.801
ECR – Avoidance	2.582	0.010 *
ECR – Anxious	−0.129	0.897
Pre-Op BES	0.631	0.528
Pre-Op PHQ-9	0.501	0.616
Pre-Op GAD-7	−0.627	0.530

ECR—Experiences in Close Relationships Scale; PHQ-9—Patient Health Questionnaire-9 Item Scale; GAD-7—Generalized Anxiety Disorder-7 Item Scale; * *p* < 0.05.

**Table 3 nutrients-11-01625-t003:** Multivariate linear regression analysis with outcome variable 2-years post-surgery Emotional Eating Scale (EES) scores (*n* = 108).

Variable	*r*	*p*
Age	0.469	0.639
Gender (Female)	−2.329	0.021 *
Race/Ethnicity (Caucasian)	0.028	0.978
Relationship Status (In a Relationship)	−0.674	0.501
ECR – Avoidance	0.407	0.686
ECR – Anxious	1.347	0.179
Pre-Op EES	0.100	0.921
Pre-Op PHQ-9	−0.821	0.412
Pre-Op GAD-7	1.241	0.216

ECR—Experiences in Close Relationships Scale; PHQ-9—Patient Health Questionnaire-9 Item Scale; GAD-7—Generalized Anxiety Disorder-7 Item Scale, * *p* < 0.05.

**Table 4 nutrients-11-01625-t004:** Multivariate linear regression analysis with outcome variable 2-years post-surgery percent total weight loss (%TWL) (*n* = 108).

Variable	*r*	*p*
Age	−1.084	0.289
Gender (Female)	−0.813	0.417
Race/Ethnicity (Caucasian)	1.177	0.247
Relationship Status (In a Relationship)	0.271	0.791
ECR – Avoidance	−1.358	0.175
ECR – Anxious	−0.286	0.778
Pre-Op Weight	0.419	0.676
Pre-Op PHQ-9	1.426	0.165
Pre-Op GAD-7	−0.686	0.494

%TWL—Percent Total Weight Loss; ECR—Experiences in Close Relationships Scale; PHQ-9—Patient Health Questionnaire-9 Item Scale; GAD-7—Generalized Anxiety Disorder-7 Item Scale.

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
