# Peer review of "Prospective Study of Attachment as a Predictor of Binge Eating, Emotional Eating and Weight Loss Two Years after Bariatric Surgery"

_nutrients, 2019, doi:10.3390/nu11071625_

Round 1
Reviewer 1 Report
Content:
This is a manuscript reporting a prospective study which examines the question whether attachment insecurity is associated with binge eating, emotional eating, and weight loss outcomes at 2-years post-surgery. This study found female gender to be a significant predictor of binge eating and emotional eating and avoidant attachment to be a significant predictor of binge eating at 2-years post-surgery.
Major comment:
From my perspective, this is a well-written manuscript. The introduction is informative and concise. The methods and the patient groups are well described and explained. The statistical analysis is appropriate, and the discussion reflects on the major findings and tries to explain the relation between avoidant attachment and binge eating.
Minor comment:
The only aspect that has not been mentioned in the manuscript is the influence
of medication. For example, the antipsychotics olanzapine and clozapine have
been shown to induce binge eating. Other medications like amphetamines like LDX
are currently used to treat binge eating disorder or tested in this indication
like GLP-1 analogues or other antidiabetic drugs.
Additionally, psychopharmacological drugs may influence your attachment. Antidepressants, for example, influence social bonding, attachment and interaction.
However, drug treatment has
not been looked at in this study. Therefore, this needs to be addressed as a
shortcoming in the discussion. If the authors have checked whether medication has
an influence on binge eating or on attachment, this should be reported, even if
the findings are negative.
Author Response
Response to Minor Comment: We appreciate the reviewer’s comments related to the effects of medication on attachment. Unfortunately, we did not collect data on medication status for patients in this study and we have thus included this in the limitations section of the manuscript.
Revisions: Discussion, Page 11, Lines 268-271

Reviewer 2 Report
In the manuscript “Prospective Study of Attachment as a Predictor of Binge Eating, Emotional Eating and Weight Loss Two Years After Bariatric Surgery” Leung et al. examine the potential relationship between insecure attachment style and eating psychopathology in patients two years post-bariatric surgery. Although this paper is interesting, several points should be addressed.
Major
1. Inclusion criteria of the Toronto Bari-PSYCH study contain life time history of psychiatric disorders (e.g. Binge Eating Disorder). Furthermore, two different types of bariatric surgery have been conducted. Please complement statistical calculation, if these factors have influence on the outcome variables of the study.
2. In the context of the Toronto Bari-PSYCH study, BES-scores, EES-scores and %TWL are collected 1-year post-surgery and 3-years post-surgery, too. Why were eating psychopathology and %TWL not examined in the course of time?
3. With reference to table 2,3 and 4: the authors name one p-value and one r-value for multivariate linear regression analysis of “Race/Ethnicity” or “Relationship Status” and the outcome variable, respectively. It is unclear, to which ethnicity or relationship status these values refer. Please specify data and add missing values.
4. Devlin et al. (reference [6], 2016) already examine eating pathology of bariatric surgery patients in a prospective study. Accordingly, statements in ll. 23 and ll. 64, that previous studies have been limited to pre-surgery samples and cross-sectional study designs, should be corrected.
5. The manuscript does not contain any visualization. Please improve the presentation of the results by adding figures.
Minor
1. The authors describe the assessment of employment status in the methods part (l. 95), but do not describe this data in later parts of the manuscript. Analysis of this variable should be added to results and discussion.
2. Please thoroughly revise the presentation of tables:
a. The legends of tables are missing and should be added.
b. Please replace variable “t” by “r” in table 2, 3 and 4 (“r” is used in the text as well).
c. With reference to table 2, 3 and 4: p-values and r-values are placed in wrong lines, which interferes with the understanding of the results.
d. Please locate the headline of table 1 left-justified.
3. In ll. 126 of the manuscript sources are missing.
4. With reference to l. 160: follow-up “1-year post-surgery” should be rectified.
5. Please revise some stylistic errors:
a. In l. 58 and l. 86 references are indicated not correctly.
b. Please switch the heading “2. Materials and Methods“ in position.
c. Often, hyphens are lacking (e.g., l. 118 and l. 122) and should be added.
d. Spelling of “2-years post-surgery“ is inconsequent and should be revised.
e. Please spell numbers in full (ll. 163).
Author Response
Response to Major Comment #1: We appreciate the reviewer’s comments related to patients’ history of psychiatric disorder and types of bariatric surgery. To clarify, not all patients who were included in this study had a history of psychiatric disorder (e.g., Binge Eating Disorder). Regarding the two types of bariatric surgery conducted, there were no significant differences between the baseline scores for the two different surgery groups.
Revisions: Results, Page 4, Lines 149-151
Response to Major Comment #2: We thank the reviewer for this feedback. Eating psychopathology and %TWL were not examined in the course of time as this was a subset of the Toronto Bari-PSYCH study that focussed on attachment and unfortunately, not all measures were available during this time.
Revisions: None
Response to Major Comment #3: We thank the reviewer for this comment. Tables 2, 3 and 4 have been updated to reflect which ethnicity and relationship status the values refer to, and the headings have been updated to “r” and “p”, as requested.
Revisions: Tables 2, 3 and 4
Response to Major Comment #4: We thank the reviewer for this feedback. Devlin and colleagues did indeed examine eating pathology of bariatric surgery patients in a prospective study. However, we are stating that attachment studies have thus far been limited to pre-surgery samples and cross-sectional designs. This is indicated in the Introduction (page 2, line 70).
Revisions: None
Response to Major Comment #5: We thank the reviewer for raising this point. We have included figures to improve the presentation of the results, as requested.
Revisions: Results, Figures 1 (a,b), 2 (a,b) and 3 (a,b)
Response to Minor Comment #1: We thank the reviewer for making note of this error. Employment was not a variable that was used in the analysis and we have removed this from the manuscript.
Revisions: Materials and Methods, Page 3, Line 102
Response to Minor Comment #2: We thank the reviewer for making note of the errors in our tables. We have placed the legends of the tables beside the respective table numbers, have replaced variable “t” to “r” in tables 2, 3 and 4, and have left-justified the headline of table 1. In regards to point c) above, we double-checked the initial submission and are unclear as to why it appeared to be misaligned.
Revisions: Results, Tables 1-4
Response to Minor Comment #3: We thank the reviewer for pointing this out. We have edited the references so that they explain all aspects of this sentence.
Revisions: Materials and Methods, Page 3, Line 128
Response to Minor Comment #4: We thank the reviewer for noting this error. We have rectified it to read “2-years post-surgery”.
Revisions: Results, Page 6, Line 178
Response to Minor Comment #5: We thank the reviewer for noting these inconsistencies. We have changed the spelling of “2-years post-surgery” to be consistent throughout the paper and spelled out numbers in full. However, in regards to points a), b), and c) above, we have double-checked the original submission and did not find these errors. It is possible that there was a formatting change or error while uploading the document. We have verified that the references are indicated correctly, that the headings are in position, and that all hyphens are in place.
Revisions:
a. None.
b. None.
c. None.
d. Abstract, Page 1, line 29; Results, Tables 2-4
e. Page 6, line 182

Round 2
Reviewer 2 Report
The authors sufficiently addressed all comments.